# Switch-Independent 3A: An Epigenetic Regulator in Cancer with New Implications for Pulmonary Arterial Hypertension

**DOI:** 10.3390/biomedicines12010010

**Published:** 2023-12-20

**Authors:** Katherine Jankowski, Vineeta Jagana, Malik Bisserier, Lahouaria Hadri

**Affiliations:** 1Center for Translational Medicine and Pharmacology, Department of Pharmacological Sciences, Icahn School of Medicine at Mount Sinai, New York, NY 10029, USA; kate.jankowski@icahn.mssm.edu; 2Cardiovascular Research Institute, Icahn School of Medicine at Mount Sinai, New York, NY 10029, USA; 3Department of Cell Biology and Anatomy & Physiology, New York Medical College, 15 Dana Road, BSB 131A, Valhalla, NY 10595, USA; vjagana@nymc.edu (V.J.); mbisseri@nymc.edu (M.B.)

**Keywords:** epigenetic, SIN3a, pulmonary arterial hypertension, cancer

## Abstract

Epigenetic mechanisms, including DNA methylation, histone modifications, and non-coding RNA, play a crucial role in the regulation of gene expression and are pivotal in biological processes like apoptosis, cell proliferation, and differentiation. SIN3a serves as a scaffold protein and facilitates interactions with transcriptional epigenetic partners and specific DNA-binding transcription factors to modulate gene expression by adding or removing epigenetic marks. However, the activation or repression of gene expression depends on the factors that interact with SIN3a, as it can recruit both transcriptional activators and repressors. The role of SIN3a has been extensively investigated in the context of cancer, including melanoma, lung, and breast cancer. Our group is interested in defining the roles of SIN3a and its partners in pulmonary vascular disease. Pulmonary arterial hypertension (PAH) is a multifactorial disease often described as a cancer-like disease and characterized by disrupted cellular metabolism, sustained vascular cell proliferation, and resistance to apoptosis. Molecularly, PAH shares many common signaling pathways with cancer cells, offering the opportunity to further consider therapeutic strategies used for cancer. As a result, many signaling pathways observed in cancer were studied in PAH and have encouraged new research studying SIN3a’s role in PAH due to its impact on cancer growth. This comparison offers new therapeutic options. In this review, we delineate the SIN3a-associated epigenetic mechanisms in cancer and PAH cells and highlight their impact on cell survival and proliferation. Furthermore, we explore in detail the role of SIN3a in cancer to provide new insights into its emerging role in PAH pathogenesis.

## 1. Introduction

Epigenetics refers to the molecular modifications of DNA that can independently regulate gene activity and, more specifically, the heritable structural and biochemical alterations of chromatin that can occur without changing the DNA sequence [1,2]. By regulating the expression of relevant genes with a local and global shift of chromatin epigenetic marks, a variety of physiological and pathological responses can also be regulated [3,4,5]. Epigenetic marks are chemical modifications of DNA and histone proteins that regulate gene expression by influencing the accessibility of specific genes to the cellular machinery responsible for transcription. These “tags” can either promote or inhibit gene expression and play crucial roles in controlling various cellular processes and developments. Recent research has focused mainly on three primary epigenetic marks, which are DNA methylation, histone modifications, and non-coding RNAs [1].

The process of DNA methylation involves the addition of a methyl group (CH3) to a cytosine base, typically occurring at CpG dinucleotides and giving rise to 5-methylcytosine (5mC) [6]. DNA methylation can repress or activate gene expression depending on the location within the gene. For example, when located in a gene promoter, DNA methylation typically represses gene transcription by impairing the binding of transcription factors and RNA polymerase to the promoter. However, methylation in the gene body is often associated with transcriptional activation and increased gene expression. Methylation can be heritable, and changes in DNA methylation can occur from one generation of cells to the next. During embryonic development and cell differentiation, dynamic processes of de novo methylation and demethylation lead to stable, cell-specific methylation patterns [6]. Histones are proteins around which DNA wraps, forming a complex called chromatin. Chemical modifications of histones can affect chromatin structure and, consequently, gene expression. Acetylation of histones usually relaxes the chromatin structure, making the DNA more accessible for transcription, thus promoting gene expression. Methylation, phosphorylation, ubiquitination, and other histone modifications can have varying effects on gene expression, depending on the specific histone and site of modification [7]. These modifications can either activate or repress transcription by regulating chromatin remodeling. Finally, non-coding RNAs (ncRNAs) such as microRNAs (miRNAs) and long non-coding RNAs (lncRNAs) interact with both DNA and RNA molecules to regulate gene expression. miRNAs can bind to messenger RNAs (mRNAs) and prevent their translation into proteins or target them for degradation [8]. lncRNAs can influence gene expression by interacting with chromatin-modifying complexes, recruiting them to specific genes, and influencing epigenetic marks. 

Epigenetic modifications are controlled by a diverse panel of epigenetic regulators, which collectively shape the gene expression landscape within cells. They are classified into three groups: writers, readers, and erasers [9]. Writers, such as DNA methyltransferases and histone methyltransferases, add epigenetic marks to DNA and histones, which can either promote or suppress gene expression. The “erasers”, like DNA demethylases and histone deacetylases, remove these epigenetic marks, allowing for dynamic adjustments in gene expression patterns. Lastly, the “readers”, like chromatin-remodeling complexes and protein domains, interpret these marks, determining the transcriptional outcomes of genes. Collectively, these epigenetic regulators ensure precise control over gene expression, which is critical for the development, function, and adaptability of cells and organisms. Over the past decade, numerous studies have demonstrated that SIN3a (Switch-insensitive 3A) is a critical component of a multiprotein complex involved in the regulation of epigenetic processes. The significance of SIN3a has received considerable attention, particularly in the field of cancer research, where its impact is broad and diverse. SIN3a has been extensively investigated in the context of various cancer types, including non-small-cell lung cancer, breast cancer, and colon cancer [10,11,12,13]. In these malignancies, SIN3a’s involvement revolves around its capacity to regulate critical aspects of tumorigenesis. It was initially described as a transcriptional co-repressor that modulates the expression of tumor suppressors and dampens the body’s natural defenses against uncontrolled cell growth [14,15]. However, recent studies have reported its novel role as a transcriptional activator in a cell- and context-specific manner. Its interactions with oncogenic factors and co-repressor complexes make it a pivotal player in cancer biology, with implications for both understanding disease mechanisms and identifying novel therapeutic targets. Research on the role of SIN3a in cancer continues to decipher the molecular complexity underlying the regulation of gene expression. Cancer and pulmonary arterial hypertension (PAH) share striking molecular similarities. Both conditions involve uncontrolled cell proliferation, similar signaling pathways, genetic mutations, and epigenetic alterations [16,17]. SIN3a’s capacity to influence gene expression in both diseases underscores its significance. Therefore, this review aims to comprehensively explore the role of SIN3a in epigenetics, particularly in the context of cancer and PAH. Understanding the role and function of SIN3a may hold the key to enhanced diagnostic and therapeutic strategies, offering a beacon of hope for patients with cancer or pulmonary vascular remodeling diseases.

## 2. Key Epigenetic Regulators

DNA methyltransferases (DNMTs) add methyl groups to the DNA molecule to transform S-adenylmethionine into 5-methylcytosine (5mC). Several DNMTs, including DNMT1, DNMT3A, and DNMT3B, have unique functions [18]. DNMT1 acts on hemimethylated DNA to create the same methylation pattern in the newly synthesized daughter strands. These proteins are associated with the maintenance of DNA methylation. Conversely, DNMT3A and DNMT3B are associated with de novo methylation and tend to bind unmethylated DNA, mainly occurring at CpG islands [18]. By adding methyl groups, transcriptional activators cannot bind easily, thus compacting the chromatin structure and repressing transcription [18]. Previous studies have shown that DNA methylation plays an important role in the regulation of superoxide dismutase 2 and a member of the TGF-β superfamily, Bone morphogenetic protein receptor type 2 (BMPR2), in the context of PAH [19,20,21]. DNMT1 downregulation is associated with passive demethylation. Another factor that tightly regulates CpG sites along with DNMT1 is the Ten-eleven translocation factor, otherwise known as TET [18]. TET enzymes mediate DNA methylation and are associated with transcription. TET proteins oxidize 5mC into new substrates, leading to active demethylation. The DNMT1 can no longer recognize the oxidized form of 5mC, so it cannot transfer any methyl groups to the DNA. Downregulation or loss-of-function mutations in TET1/2 have been found in several cancers and are associated with uncontrolled proliferation and tumorigenesis [22]. TET2 is significantly downregulated in most patients with PAH and is associated with an increase in inflammation [23].

Along with DNA modifications, histone modifications can also modulate gene expression levels. A major factor that regulates histone structures are the histone acetyltransferases (HATs) and histone deacetyltransferases (HDACs) which remove acetyl groups from the histone proteins [7]. Acetylation neutralizes the positive charge of lysine and can weaken the interactions between histones and DNA, contributing to a more open chromatin state [7]. Several members of the HDAC group have been identified in patients with cancer and PAH. P300, a histone acetyltransferase (HAT), is usually recruited to transcriptional enhancers to modulate gene expression by acetylating histones. Acetylation of p300/CBP occurs at lysine 27 within the regulatory loop of its acetyltransferase domain, likely through a highly efficient and cooperative intermolecular reaction, and enhances its acetyltransferase activity and binding to proteins [24,25]. Histone acetylation is recognized by bromodomains, which act as “readers”. The bromodomain and extra-terminal (BET) family proteins contain bromodomains that recognize acetyl groups on histones and serve as transcriptional co-activators [26]. They are involved in processes such as cell-cycle progression, transcriptional activation, and elongation. One essential BET protein, bromodomain-containing protein 4 (BRD4), binds to acetylated histones and various transcription factors via an acetylation-dependent mechanism [27]. BRD4 acts as a framework for transcription factors in promoters and super-enhancers. This interaction plays a role in the expression of oncogenes, pro-inflammatory cytokines, and chemokines, making it a valuable target for the treatment of diseases such as inflammation and cancer. Inhibitors targeting bromodomains have been explored as potential therapeutic interventions using small-molecule compounds and are currently under investigation in clinical trials for treating cancer and PAH [28]. 

Histone methylation is critical for regulating chromatin structure by affecting DNA compaction and accessibility, leading to changes in gene expression and overall cellular function. Enhancer of zeste homolog 2 (EZH2) is a histone methyltransferase belonging to the Polycomb Repressive complex 2. EZH2 catalyzes the trimethylation of histone H3 at lysine 27 (H3K27me3) and is frequently upregulated in human cancers [29]. It plays a crucial role in regulating histone methylation and influencing chromatin structure. This modification is commonly associated with the repression of gene expression, thereby affecting cellular processes such as development, differentiation, and disease [29]. Gain-of-function mutations and increased EZH2 expression have been previously identified in patients with ovarian cancer, non-small-cell lung cancer, melanoma, breast cancer, prostate cancer, pancreatic cancer, and hematologic malignancies [29].

Recently, several studies have revealed the significant role of Switch-insensitive 3A (SIN3a) as a crucial component in the regulation of epigenetic processes. SIN3a interacts with other proteins, such as EZH2 and HDACs, to regulate gene expression. These interactions contribute to a dynamic epigenetic landscape and directly affect disease development and progression. Understanding the role of SIN3a in regulating gene expression through epigenetic mechanisms is essential for deciphering the underlying factors contributing to diseases (Figure 1).

## 3. Epigenetic Regulator SIN3a

### 3.1. Molecular Function of SIN3a

SIN3 was initially identified at the end of the twentieth century as a global transcriptional regulator [30]. In mammals, the large scaffold protein complex has two isoforms, SIN3a and SIN3b, which form different complexes [31]. Previous studies have also determined that heterozygous mutations or deletions in each SIN3 complex (SIN3a and SIN3b) resulted in syndromic genetic defects, while additional animal models with SIN3a deletions died during embryogenesis [32], all of which suggest that the SIN3a complex is vital for early embryonic and standard development [21,31]. The SIN3 complex, through prior studies, was determined to be a large scaffold protein that contains HDAC1 and HDAC2, the SIN3a and SIN3b corepressors, and sequence-specific transcriptional regulators, all contained within four amphipathic helix domains and an HDAC interaction domain [15,30,33,34,35]. 

Bansal et al. thoroughly described how two paralogs of the mammalian Sin3 family, SIN3a and SIN3b, have overlapping yet separate roles, where SIN3a and SIN3b are believed to have comparable scaffolding capacities [35,36]. SIN3a and SIN3b interact with transcription factors such as p53, Mad1, KLF, REST, and ESET, while SIN3b exclusively interacts with specific transcription factors like CIITA [34,35,36]. In this study, the authors investigated the roles of SIN3 proteins in muscle development and found that reduced levels of SIN3a were associated with sarcomere structure issues in myotubes [37]. The SIN3b deletion increased SIN3a recruitment at target loci; however, the absence of a corresponding reverse effect indicates the distinct functional nature of the SIN3a and SIN3b complexes [36,37]. Although SIN3a does not specifically bind to DNA, repressed gene expression may occur when the scaffold allows for the assembly of transcriptional factors that promote the recruitment of HDACs [21]. According to numerous studies, SIN3a stimulates the transcription of specific target genes, while furthermore, it has been determined that SIN3a forms complexes with additional regulatory proteins to silence the transcription of several cycle-regulating and tumor-progression genes [38,39,40,41,42]. 

### 3.2. Interactions with Histone Deacetylases (HDACs) and Other Partners

Dave et al. recently published a comprehensive review of histone deacetylases (HDAC) by examining their pathways throughout research history. HDACs are extremely relevant to PAH studies. Previous studies have shown a significant increase in the protein expression of HDAC1 and HDAC5, both of which are found in human lung tissue from idiopathic PAH patients and in the right ventricle (RV) and lungs of animal models of hypoxia-induced PAH [19,43]. Additionally, previous research has also shown elevated levels of histone acetylation of H3 and H4 in the promoter region of the eNOS gene in the pulmonary arterial endothelial cells of infants with persistent pulmonary hypertension of the newborn [19,44]. As recent studies have emphasized the important role of HDACs in RV failure in PAH experimental models, targeting histone acetylation represents an emerging area of research with promising benefits for understanding the development of PAH and generating novel therapeutic strategies [19]. Cavasin et al. evaluated histone deacetylase inhibition by exposing rats to a hypoxic environment in the presence or absence of a benzamide HDAC inhibitor, MGCD0103, which inhibits HDACs 1, 2, and 3 [45]. The inhibitor compound reduced pulmonary arterial pressure and improved the pulmonary arterial acceleration time. These results were then compounded with a similar independent class I HDAC-selective inhibitor, MS-275, where the right ventricular function was ultimately maintained in the animals treated with MGCD0103 [45]. Several studies have pointed to the fact that while selective HDAC inhibitors have shown promising results in preclinical studies, the use of such inhibitors is linked to debilitating side effects [46,47,48]. The SIN3/HDAC co-repressor complex regulates gene suppression via chromatin condensation. SIN3, on account of its association with DNA-binding proteins, serves as a central support structure that assembles histone deacetylases (specifically class I HDACs, including HDAC1 and HDAC2) and additional enzymes involved in modifying chromatin at specific gene promoters [30,35] Conventionally, the SIN3/HDAC complex mediates the deacetylation of histones H3 and H4. When coupled with the dynamic regulation of methylation and demethylation of specific histone lysines, this process is instrumental in achieving the gene repression facilitated by SIN3 [13,35,37,49]. In a recent study, our group tested the effect of SIN3a overexpression on HDAC1/2 activity [21]. We demonstrated that SIN3a overexpression decreased HDAC1/2 activity in human pulmonary arterial smooth muscle cells (PASMCs) and that the knockout of SIN3a with specific small interfering RNA ultimately increased HDAC1/2 activity and altered the regulation of genes of interest in PASMCs [21]. 

### 3.3. Role of SIN3a in Transcriptional Regulation and Chromatin Remodeling: Co-Repressor and Activator

SIN3a is a scaffold protein and co-repressor that plays a vital role in the maintenance of chromatin structure and regulation of gene transcription but has also been linked to aberrant gene regulation in cancer and cardiovascular diseases [14]. Increasing evidence suggests that SIN3a is a protein with dual functions as an activator and repressor of gene expression in cellular or promoter contexts. Studies have specifically determined that deletion of SIN3 is associated with the downregulation and upregulation of transcripts [50,51]. Our recent study also confirmed that SIN3a was significantly downregulated in the lung samples of human PAH patients and animal models of PAH, and the restoration of SIN3a in PASMCs inhibited cell growth in vitro and animal models of PAH [21] (Figure 2).

## 4. Emerging Roles of Epigenetic Regulator SIN3a in Cancer

Indications for an epigenetic link to cancer were derived from gene expression and DNA methylation studies [52]. Studies regarding this epigenetic link are vast and cover a great range of history, as detailed by Feinberg and Tycko’s review of the history of cancer genetics [53]. While many initial studies purely drew correlations between cancer and genetics, a few studies highlighted in this review demonstrated a connection between epigenetic pathways and cancer [52]. As more research has emerged, new studies performed by the International Cancer Genome Consortium have significantly strengthened the early observed link between epigenetic pathways and cancer, and whole-genome sequencing in a wide variety of cancers has allowed an accessible catalog of somatic mutations in epigenetic regulators to be made available [54,55].

### 4.1. SIN3a’s Involvement in Cancer Development and Progression

A study performed by Ellison-Zelski et al. previously determined that SIN3a is expressed in breast cancer cells, and they further expanded their studies to conclude that SIN3a is a regulator of the gene expression, growth, and survival of estrogen receptor alpha-positive breast cancer cells [56]. Other studies also determined that SIN3a promotes cell growth and survival [15,57]. The results of these studies suggest that SIN3a primarily functions as a transcriptional repressor and that gene repression is comparable to gene activation as a vital determinant of cell growth [15,56,57]. The specific finding of SIN3a as a pro-survival factor highlights the importance of the estrogen-mediated survival of breast cancer cells, where SIN3a knockdown increases apoptosis without affecting the cell cycle [56]. This further supports previous studies and findings regarding estrogen-mediated repression of apoptosis, which has an incredible impact on the overall growth of cells [58,59].

### 4.2. SIN3a’s Impact on Oncogenes and Tumor Suppressor Genes

Bansal et al. provided an excellent description of the SIN3 family, and presented a breakdown of the paralogs SIN3a and SIN3b, and thoroughly described the impact of SIN3a on oncogenes and tumor suppressor genes. Under conditions of cellular stress, SIN3a protein expression increases [36]. Under genotoxic stress, there is a significant increase in levels of both SIN3a and SIN3b, contributing to the stability and trans-repressive functions of the TP53 tumor suppressor [35,49,60,61,62]. The SIN3 proteins interact with the retinoblastoma family of tumor suppressors through the retinol-binding protein 1 (RBP1). This interaction represses the transcription of E2F-responsive pro-proliferative genes [35,63]. Furthermore, RBP1 engages with BRMS1, a protein identified for its ability to inhibit metastasis in various cancer types [36,60,61,62]. The authors suggested that genetic manipulations are a classical method for identifying gene function in tumorigenesis. Although techniques such as gene deletions, overexpression, or RNA interference have led to innovative discoveries, their use varies from protein to protein, wherein manipulations like deleting SIN3a, in comparison to interfering with specific protein interactions of SIN3 proteins, can affect the function of the residual complex [13,36,63]. 

### 4.3. Mechanisms of SIN3a-Mediated Epigenetic Alterations: Examples in Different Types of Cancer

In their study regarding breast cancer, Li and collaborators evaluated the pathophysiological function and underlying mechanism of FOXN3, a forkhead transcriptional repressor that is physically associated with the SIN3a complex in ER+ breast cancer cells [64]. The authors analyzed the genomic targets of a complex containing FOXN3, NEAT1, and SIN3a, and further identified a panel of genes that are vitally involved in the epithelial-to-mesenchymal transition [64]. Many previous studies have also determined multiple explanations regarding the mechanistic role of SIN3a, including one by Gambi et al., who evaluated STAT3-addicted tumors and determined that SIN3a is a regulator of STAT3, while also identifying the STAT3/SIN3a axis as a possible therapeutic target [65]. Lewis et al. generated stable knockdown cells of SIN3 paralogs, both individually and in combination, using three distinct non-overlapping small hairpin RNAs. In this study, the SIN3b knockdown decreased transwell invasion and the number of invasive colonies, while the knockdown of SIN3a increased transwell invasion and the number of invasive colonies [10]. Their results were corroborated in vivo, where the SIN3b knockdown decreased lung metastases and the SIN3a knockdown increased lung metastases [10]. 

### 4.4. Targeting SIN3a as a Potential Therapeutic Approach in Cancer

Epigenetic modifiers regulate gene expression and often form large multiprotein complexes [36,66]. Targeting dysregulated epigenetic writers or erasers represents a compelling therapeutic approach in cancer because of the reversibility of chromatin structure [36]. Ongoing research aims to develop and identify novel inhibitors, like azacitidine and suberoylanilide hydroxamic acid, that target chromatin-associated proteins [35]. 

A study by Yao et al. showed that SIN3a regulates the fate of epithelial progenitor cells during lung development. Using SIN3a knockout mice and SIN3a loss-of-function (LOF) in epithelial cells, the study showed that SIN3a loss not only replicated many of the deficiencies seen in cases of HDAC1/2 deficiency but also induced more profound abnormalities during the initial phases of lung development [67]. The authors determined that the sonic hedgehog (Shh)^Cre^ driver line was employed to facilitate efficient recombination of the respective floxed alleles within the developing foregut endoderm, suggesting two potential scenarios: first, that the SIN3a-containing HDAC1/2 complex plays a predominant role in mediating HDAC1/2-dependent effects on lung development, or that SIN3a performs HDAC1/2-independent functions that influence the fate of the early lung endoderm [67]. The limited overlap observed between the altered lung transcriptomes of SIN3a LOF and HDAC1/2 LOF embryos can be attributed to the fact that the former exhibits much earlier developmental anomalies [67].

The historical implication that cellular senescence is linked to the pathogenesis of numerous chronic pulmonary ailments raises the possibility that SIN3a could potentially influence the regulation of epithelial progenitor cell destinies within the postnatal lung [67]. The authors concluded that subsequent investigations are warranted to elucidate SIN3a’s functional involvement in these afflictions and ascertain whether analogous transcriptional regulatory mechanisms govern cell fate determination in the postnatal pulmonary context. However, these studies shed light on the pivotal role of specific co-repressor proteins, including SIN3a, in orchestrating tissue-specific transcriptional dynamics and pivotal cell-fate determination during the intricate process of lung development [67]. 

A previous study showed that SIN3a is reduced in 19 of 31 cases (61%) of non-small-cell lung cancers, and its absence may alter the expression of growth-related genes in an epigenetic-dependent manner through histone acetylation, leading to tumorigenesis in lung cancer cells [68]. There are remarkable similarities between PAH and cancer, including hyperproliferation, resistance to apoptosis, and dysregulation of tumor suppressor genes.

## 5. The Cancer-like Theory of PAH

PAH is characterized by an increased proliferative state and apoptosis resistance, affecting pulmonary vascular cells, including pulmonary artery smooth muscle cells, endothelial cells, and fibroblasts. Ultimately, this leads to extensive vascular remodeling and obstruction of the distal pulmonary arteries. At the molecular level, these cells exhibit characteristics that are similar to those of cancer cells. Many growth factors and inflammatory markers have been identified in the development of both cancer and PAH [69,70]. These inflammatory molecules are directly linked to the proliferation of pulmonary arterial cells. Importantly, previous studies revealed specifically increased levels of IL6 in PAH patients that were correlated with worsened prognoses [71]. For example, IL-6 knockout mice were protected against chronic hypoxia-induced PAH, whereas IL-6 overexpressing mice developed spontaneous PAH [72]. Other factors have similar effects by activating subsequent signaling pathways, such as MEK/ERK, PI3K/AKT, and JAK/STAT3, resulting in increased cell growth, cell-cycle progression, and cell survival [73]. Additionally, STAT3 has been extensively investigated in cancer and PAH [74,75,76,77,78]. Similar to gene mutations that predispose patients to certain types of cancer, BMPR2 is a common risk factor for PAH development. Patients with BMPR2 mutations or loss of BMPR2 expression and/or function tend to develop PAH at a younger age and are often associated with a worse prognosis [79,80,81,82,83,84,85]. Recently, epigenetic modulators were also seen to play a profound role in PAH onset by affecting all of the main epigenetic layers, such as DNA methylation levels, histone modifications, and non-coding RNAs such as long non-coding RNA and micro-RNAs. These different layers of epigenetic regulation profoundly affect gene expression levels, increase cell survival rates, and decrease apoptosis in PAH, further emphasizing the importance of understanding how they are tightly regulated and may contribute to the complexity and heterogeneity of PAH pathogenesis (Figure 3).

## 6. New Insights into the Roles of SIN3a in PAH

Pulmonary hypertension (PH) is characterized by remodeling of the pulmonary vasculature, hypertrophy, and remodeling of the right ventricle, and it is defined as a mean pulmonary arterial pressure greater than 20 mmHg at rest, as recorded by right heart catheterization [86]. PAH refers to Group 1 of PH and is classified as a relatively rare form of pulmonary hypertension, including idiopathic PAH, heritable PAH, and other forms of PAH associated with diseases or conditions like scleroderma, schistosomiasis, lupus, congenital heart disease, chronic liver disease, HIV, drugs, or toxins [19,87]. Han et al. thoroughly cover the basics of PAH and cancer metabolism, where PAH is defined as a vascular disease that is characterized by high pressure in the pulmonary artery, which leads to right heart failure and premature death [88]. PAH patients’ lungs contain plexogenic lesions with hyperproliferating cells, apoptosis-resistant pulmonary arterial endothelial cells (PAECs), and fibroblasts, while additionally displaying intimal and medial thickening of the pulmonary arteries [88]. 

Recently, Bisserier et al. primarily focused on the SIN3 complex in PAH, in the context of whether high levels of *BMPR2* DNA methylation were correlated with changes in SIN3 expression in PAH [21]. Using real-time quantitative polymerase chain reactions and Western blot analysis, the group determined a fundamental finding that the expression levels of SIN3a mRNAs and proteins in PAH-patient lung samples are significantly decreased compared to those of non-PAH-patient lung samples [21]. The group further analyzed the expression of BMPR2 and SIN3a in both a Sugen–chronic hypoxia-induced PAH (SuHx) model in mice and a monocrotaline (MCT)-induced PAH model in rats [20]. A consistent link emerged between BMPR2 and SIN3a mRNA and protein levels in animal models, in contrast to control groups. This prompted the determination that the reduction in SIN3a expression in PAH is a common feature observed across several animal species [20]. In this study, dual approaches were used: siRNA SIN3a for the knockdown of SIN3a expression and a lentivirus-mediated overexpression for a gain-of-function strategy [20]. The findings hold significance, given that vascular remodeling in PAH is marked by increased proliferation and migration of pulmonary vascular cells [21]. The results from this study revealed that the depletion of SIN3a significantly increased PASMC proliferation and migration, while the overexpression of SIN3a had the opposite effect [21]. Contrastingly, the overexpression of SIN3a increased *BMPR2* mRNA and protein expression in PASMCs and decreased the marker of proliferation CCND1, which suggests that the loss of SIN3a expression may potentiate PASMC proliferation in a BMPR2-dependent manner [21]. SIN3a overexpression in PASMCs decreased *BMPR2* promoter methylation [21]. 

### 6.1. SIN3a Molecular Mechanisms in PAH

McDonel et al.’s findings highlighted the importance of SIN3a in embryonic development, as evidenced by the lethality observed in SIN3a knockout mice [31]. Bisserier et al. demonstrated that restoring SIN3a expression in proliferating PASMCs leads to the modulation of various gene expressions. Notably, the loss of SIN3a in PAH promoted EZH2-mediated histone methylation (H3k27me3) and DNA methylation in the *BMPR2* promoter region, which compromised the transcriptional machinery [20], which ultimately repressed BMPR2 expression in PAH [21]. This study also showed that SIN3a deceased HDAC1 activity and decreased histone methylation by reducing the enrichment of H3K27me3 at the *BMPR2* promoter region by antagonizing the expression of EZH2 and also demonstrated that SIN3a increased TET1-mediated DNA demethylation of the *BMPR2* promoter region, thus identifying the SIN3a/EZH2-H3K27me3/TET1 pathway axis as a changing epigenetic mechanism that involves *BMPR2* expression and vascular remodeling in PAH [21]. Ultimately, their results suggested that hypermethylation of the *BMPR2* promoter region is associated with decreased BMPR2 expression in human patients with PAH, and SIN3a dynamically regulates BMPR2 DNA methylation and expression by modulating the EZH2 and balancing DNMT1/TET1 expression [21].

### 6.2. Comparison of SIN3a’s Effects on Cancer and Its Implications for PAH

The role of EZH2 in carcinogenesis has been extensively documented in numerous studies showing that EZH2 upregulation, or gain-of-function mutation, is often correlated with an adverse prognosis in various cancer types, including melanoma, prostate cancer, breast cancer, and others [21,89,90,91,92,93,94,95,96,97,98]. Several cancer cell lines proliferate when EZH2 is overexpressed, whereas tumor growth is inhibited by EZH2 suppression. Surprisingly, EZH2 has been demonstrated to prevent the progression of PAH by preventing hPASMC migration and proliferation [21,99]. The potential benefits of EZH2 inhibitors as cancer therapies are highlighted by the fact that inhibiting EZH2 decreases the proliferation of numerous cancer cell lines [100]. Interestingly, another study showed that human lung samples from PAH patients exhibited increased EZH2 expression [21]. 

### 6.3. Exploring SIN3a as a Novel Therapeutic Target for PAH

A study by Bisserier et al. evaluated the therapeutic efficacy of lung-targeted gene transfer of SIN3a in an MCT-induced PAH rat model, where rats were initially subjected to MCT-induced PAH and then treated with intratracheal delivery of either aerosolized AAV1.hSIN3a or AAV1.Luc as a control [20]. The intratracheal delivery demonstrated specificity towards the lungs, as evidenced by a significantly higher number of exogenous SIN3a genome copies in the lungs of the AAV1.hSIN3a-treated group compared to the control group [20]. The mRNA and protein expression of SIN3a was effectively restored in the lungs, even influencing the endogenous SIN3a transcript of the rats [20]. The therapeutic impact of AAV1.hSIN3a was further confirmed by improvements in pulmonary artery and RV pressures, reduced distal pulmonary vascular remodeling, and attenuated RV hypertrophy [20]. Notably, AAV1.hSIN3a treatment decreased the mean pulmonary arterial pressure, mitigated the adverse hemodynamic profiles, and reduced RV hypertrophy [20]. These findings indicate lung-targeted instillation of AAV1.hSIN3a holds promise for treating PAH [21]. 

The group also evaluated the therapeutic efficacy of AAV1.hSIN3a gene therapy in preclinical Sugen/Hypoxia induced-PAH in mice, a commonly used model to assess the efficiency of new therapies in PAH [21]. Two distinct protocols were implemented: the treatment entailed the administration of AAV1.hSIN3a post-PAH induction and prevention involved the delivery of AAV1.hSIN3a before PAH initiation [21]. The gene therapy led to the expression of human SIN3a in the pulmonary tissue and the restoration of endogenous SIN3a levels in the mice, and prolonged SIN3a expression effectively mitigated the muscularization and remodeling of small pulmonary arteries, as well as RV remodeling [21]. RV remodeling was also ameliorated, cardiomyocyte hypertrophy was decreased, and the expression levels of markers associated with cardiac hypertrophy were reduced [21]. Furthermore, the restoration of SIN3a had a broader impact on gene expression, resulting in the upregulation of TET1, ELP3, and MBD4 and the downregulation of DNMT1, EZH2, SUZ12, and CTCF, which led to a reduction in the methylation level of the BMPR2 promoter region and the re-establishment of BMPR2 expression in the animal models of PAH [20]. Immunoblot analysis corroborated increased levels of TET1, BMPR2, and pSMAD1/5/9, while levels of EZH2, DNMT1, CTCF, and CCND1 were diminished in the AAV1.hSIN3a-treated group compared to controls [21].

## 7. Conclusions

Due to SIN3a’s role in regulating epigenetic modifications, especially in cancer processes, and as many pathways underlying cancer development have been identified in PAH, SIN3a has become a key topic in PAH research. As SIN3a is a scaffold protein, it recruits many activators and/or repressors of gene transcription to different sites, adjusting the levels of expression of certain genes and causing changes in downstream effects, such as alterations in cell proliferation and growth. Notably, SIN3a is not gene-specific and can modulate the expression of a wide array of genes. Since SIN3a has a dual role as an activator and repressor of transcription, its effects are mainly dependent on the cell type and the environment in which it functions. Thus, SIN3a activity can change depending on the context, including cell type, stressors, environment, and the presence of cofactors. Additionally, to study the effect of SIN3a on gene expression, adjustments to SIN3a activity levels must be fully understood to obtain a more comprehensive picture of SIN3a’s broad effects. Unfortunately, only SIN3a inhibitors are currently available for use right now. Because of the diverse effects of SIN3a and its changes in activity depending on the cell in which it functions, more cell types should be studied to highlight and identify its function in the context of health and disease. This may ultimately lead to a better understanding of its capacity to act as a transcriptional regulator. Additionally, the development of SIN3a activators will provide further information on the extent of its impact on cellular pathways. Future approaches may involve the application of CRISPR technology to precisely modify the epigenetic landscape of a specific gene in a targeted manner, thereby minimizing off-target effects. This can be achieved by employing tissue-specific promoters designed for specific cell types for gene therapy. Another emerging perspective entails the development of pharmacological activators for PAH treatment.

## Figures and Tables

**Figure 1 biomedicines-12-00010-f001:**
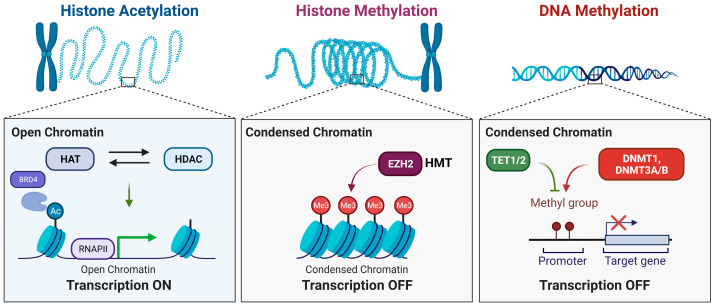
Major epigenetic pathways. Epigenetic modifications occur in DNA and histone proteins to regulate gene expression by adjusting chromatin structure and accessibility. The three main epigenetic marks are histone acetylation/methylation and DNA methylation. Histone acetylation is mainly regulated by histone acetyltransferases (HATs) and histone deacetyltransferases (HDACs), which add and remove acetyl groups, respectively, from histones. Acetylation leads to the opening of chromatin, making it easier for transcription to occur at these sites. Histone methylation occurs through the enhancer of zeste homolog 2 (EZH2), which catalyzes the tri-methylation of lysine 27 on histone 3. H3K27me3 is associated with gene repression. Finally, DNA methylation is regulated by DNA methyltransferases (DNMTs) and Ten-eleven translocation (TET) enzymes, which add and remove methyl groups on the DNA, respectively. DNA methylation prevents the binding of transcription factors. Created with BioRender.com, accessed on 18 December 2023.

**Figure 2 biomedicines-12-00010-f002:**
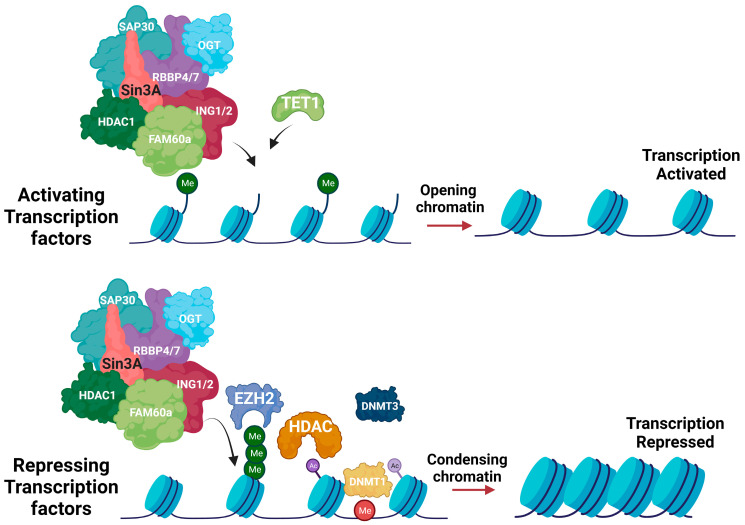
SIN3a is a transcriptional activator and repressor. SIN3a is a scaffold protein that can recruit different transcription factors to the sites of transcription, either activating or repressing gene expression according to the factors that are being recruited. SIN3a recruits both repressing and activating factors in a cell-, tissue-, and context-specific manner. If it recruits histone, DNA methyltransferase, or histone acetylase, the chromatin will be condensed, repressing transcription. If it recruits histone acetylases, histones, or DNA demethyltransferases, chromatin will be in a decondensed state, allowing transcriptional activation. Created with BioRender.com, accessed on 18 December 2023.

**Figure 3 biomedicines-12-00010-f003:**
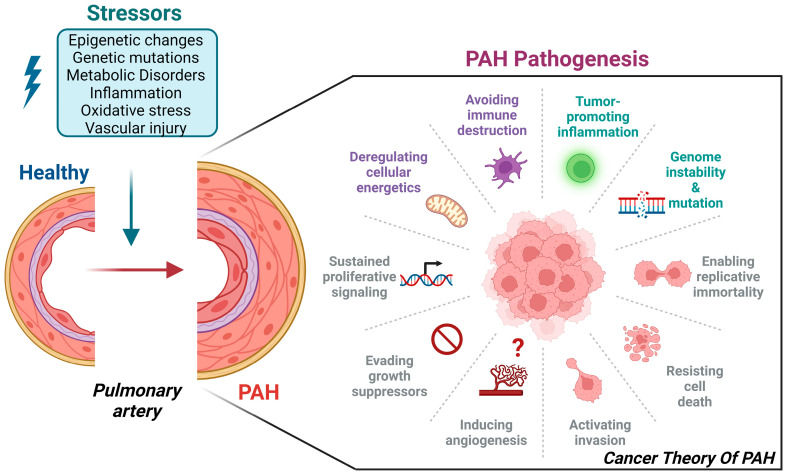
The cancer theory of PAH. PAH and cancer share many similarities in pathogenesis. Several of the main characteristics of cancer, such as cell-death resistance, sustained and increased cell proliferation, and genomic instability, are also key factors in PAH. Increased stresses, such as inflammation, vascular injury, genetic mutations, or the environment, are major biological processes in the pulmonary artery that contribute to vascular remodeling and progressive obstruction of the distal pulmonary arteries during PAH pathogenesis. Created with BioRender.com, accessed on 18 December 2023.

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
