# Peer review of "Switch-Independent 3A: An Epigenetic Regulator in Cancer with New Implications for Pulmonary Arterial Hypertension"

_biomedicines, 2023, doi:10.3390/biomedicines12010010_

Round 1
Reviewer 1 Report
Comments and Suggestions for Authors
The manuscript titled “SIN3a: An Epigenetic Regulator in Cancer with New Implications for Pulmonary Arterial Hypertension" explores the role of SIN3a, a scaffold protein involved in epigenetic regulation, in both cancer and pulmonary arterial hypertension (PAH). SIN3a influences gene expression by modulating transcriptional factors that modify epigenetic marks, thereby impacting critical biological processes like cell proliferation, differentiation, and apoptosis. This regulation is particularly significant in the context of tumor suppressors and oncogenes. SIN3a's interactions with histone deacetylases (HDACs) highlight its relevance in PAH, where increased levels of HDAC1 and HDAC5 have been observed in iPAH patients. These findings suggest that targeting histone acetylation could be a promising approach in PAH treatment strategies​.
In cancer, SIN3a functions primarily as a transcriptional repressor, influencing gene expression, cell growth, and survival. This role is evident in various cancer types, including breast cancer​​. Drawing parallels between cancer and PAH, the study notes similarities in cell proliferation and apoptosis resistance. This comparison opens avenues for applying cancer therapeutic strategies in PAH​​.
I read this manuscript thoroughly, the manuscript is well-written and organized. Only one issue needs a correction, in ABBREVIATIONS part, 5mc should be 5mC. This is a fantastic review.
Author Response
We sincerely thank the editors and reviewers for reviewing our manuscript thoroughly. We have revised the manuscript based on these comments, and detailed responses to each reviewer’s comments can be found below. We have included the revised sections in red font to indicate the changes. Additionally, we have addressed each reviewer’s comments individually, providing detailed responses below their respective comments. We would like to thank the reviewers for their insightful comments. Their contributions have been invaluable in enhancing the quality of the manuscript.
Reviewer #1
The manuscript titled “SIN3a: An Epigenetic Regulator in Cancer with New Implications for Pulmonary Arterial Hypertension" explores the role of SIN3a, a scaffold protein involved in epigenetic regulation, in both cancer and pulmonary arterial hypertension (PAH). SIN3a influences gene expression by modulating transcriptional factors that modify epigenetic marks, thereby impacting critical biological processes like cell proliferation, differentiation, and apoptosis. This regulation is particularly significant in the context of tumor suppressors and oncogenes. SIN3a's interactions with histone deacetylases (HDACs) highlight its relevance in PAH, where increased levels of HDAC1 and HDAC5 have been observed in iPAH patients. These findings suggest that targeting histone acetylation could be a promising approach in PAH treatment strategies​.
In cancer, SIN3a functions primarily as a transcriptional repressor, influencing gene expression, cell growth, and survival. This role is evident in various cancer types, including breast cancer​​. Drawing parallels between cancer and PAH, the study notes similarities in cell proliferation and apoptosis resistance. This comparison opens avenues for applying cancer therapeutic strategies in PAH​​.
I read this manuscript thoroughly, the manuscript is well-written and organized. Only one issue needs a correction, in ABBREVIATIONS part, 5mc should be 5mC. This is a fantastic review.
Response: We sincerely appreciate your careful review of our revised manuscript and your thoughtful suggestions on how to improve it. Your comments about the overall quality, organization, and content of our review are encouraging, and we are grateful for your positive assessment. In response to the reviewer’s suggestion, we have promptly replaced "5mc" with “5mC” in the ABBREVIATIONS section as well as in the manuscript.
Reviewer 2 Report
Comments and Suggestions for Authors
This is a review article on the possible involvement of SINA3 in PAH. However, there still exists a lot of problems in the article.
1. There are a lot of language problems in the article, so it needs in-depth review.
2. “SIN3a acts as a scaffold protein and was initially 16 described in cancer as a major regulator of epigenetics.” Should be changed to “SIN3a acts as a scaffold protein and was initially 16 described in cancer as a major epigenetic regulator”.
3. In sentence “However, activation or 20 repression of gene expression…”, “the” should be added to the place before “activation”
4. In sentence “…gene expression depends on what factors interact with SIN3a…”, “what” should be changed to “the”.
5. In sentence “This comparison offers new therapeutic possibilities”, “possibilities” should be changed to “options”.
6. For language, et al.
7. In the summary section, the overall summary is too long and not concise enough. This paper mainly discussed the relationship between SINA3 and PAH, but in the review, the author spent a lot of space describing the relationship between SINA3 and tumor. We recommend the authors to review the section of abstract.
Comments on the Quality of English Language
There are a lot of language problems in the article, so it needs in-depth review.
Author Response
We sincerely thank the editors and reviewers for reviewing our manuscript thoroughly. We have revised the manuscript based on these comments, and detailed responses to each reviewer’s comments can be found below. We have included the revised sections in red font to indicate the changes. Additionally, we have addressed each reviewer’s comments individually, providing detailed responses below their respective comments. We would like to thank the reviewers for their insightful comments. Their contributions have been invaluable in enhancing the quality of the manuscript.
Reviewer #2:
- There are a lot of language problems in the article, so it needs in-depth review.
Response: In response to the reviewer’s comment regarding language problems in the article, we appreciate the feedback and completely agree with the importance of ensuring the manuscript’s language is clear and comprehensible. The entire manuscript was thoroughly reviewed by two experienced native English-speaking colleagues to address language issues, including grammar, syntax, and overall readability. We have worked diligently to enhance the clarity and coherence of the text to improve the manuscript’s language quality further.
- “SIN3a acts as a scaffold protein and was initially 16 described in cancer as a major regulator of epigenetics.” Should be changed to “SIN3a acts as a scaffold protein and was initially 16 described in cancer as a major epigenetic regulator”.
Response: Thank you for your suggestion. we have indeed replaced the original sentence with a more detailed explanation that encapsulates SIN3a's function as a molecular platform. The revised sentence now reads: “SIN3a serves as a scaffold protein and facilitates interactions with transcriptional epigenetic partners and specific DNA-binding transcription factors to modulate gene expression by adding or removing epigenetic marks”. This revision is intended to provide a clearer and more comprehensive description of SIN3a's role in epigenetic regulation, aligning with the focus of our review on its implications in both cancer and PAH. Please see abstract page 1, lines 13-15.
- In sentence “However, activation or repression of gene expression…”, “the” should be added to the place before “activation”
Response: Thank you for pointing that out. We have corrected the manuscript as follows: “However, the activation or repression of gene expression...”. Please see page 1, lines 15-17.
- In sentence “…gene expression depends on what factors interact with SIN3a…”, “what” should be changed to “the”.
Response: We sincerely apologize for the oversight in the sentence mentioned and agree that the sentence should be revised for improved clarity. We have corrected the sentence as suggested, changing “what” to “the”. “Gene expression depends on the factors that interact with SIN3a...” Please see page 1, line 16.
- In sentence “This comparison offers new therapeutic possibilities”, “possibilities” should be changed to “options”.
Response: We appreciate the reviewer’s careful attention to the wording of the manuscript. We acknowledge the suggestion to replace “possibilities” with “options” for greater clarity and have now incorporated this change accordingly: “This comparison offers new therapeutic options.” Please see page 1, lines 26.
- For language, et al.
Response: We have thoroughly reviewed our manuscript and made the necessary adjustments to our in-text citations. Specifically, we have appropriately used "et al." after the first author's name to maintain consistency and readability, as well as to adhere to standard academic writing practices.
- In the summary section, the overall summary is too long and not concise enough. This paper mainly discussed the relationship between SINA3 and PAH, but in the review, the author spent a lot of space describing the relationship between SINA3 and tumor. We recommend the authors to review the section of abstract.
Response: We appreciate your feedback. In response, we have carefully revised the abstract to align it more closely with the manuscript’s primary focus on the relationship between SIN3a and cancer, while providing new perspectives in the field of PAH. Importantly, we have condensed the abstract and further highlighted the translational potential of this cancer research in understanding PAH, offering new insights into the role of SIN3a in the epigenetic regulation in pulmonary vascular cells and its implications in PAH. We believe this revised abstract now concisely and effectively communicates the overall objective of our manuscript.
Reviewer 3 Report
Comments and Suggestions for Authors
This is a comprehensive review of currently available evidence on epigenetic mechanisms in relation to cancer and pulmonary arterial hypertension, with special emphasis on SIN3a. The manuscript is well written, scientifically sound and interesting. I would suggest adding few lines on DNA methylation of regions other than promoters, as the effects usually differ. Also, both PAH and PH were used to denote pulmonary arterial hypertension, and the authors should stick to only one throughout the text.
Author Response
We sincerely thank the editors and reviewers for reviewing our manuscript thoroughly. We have revised the manuscript based on these comments, and detailed responses to each reviewer’s comments can be found below. We have included the revised sections in red font to indicate the changes. Additionally, we have addressed each reviewer’s comments individually, providing detailed responses below their respective comments. We would like to thank the reviewers for their insightful comments. Their contributions have been invaluable in enhancing the quality of the manuscript.
Reviewer #3:
This is a comprehensive review of currently available evidence on epigenetic mechanisms in relation to cancer and pulmonary arterial hypertension, with special emphasis on SIN3a. The manuscript is well written, scientifically sound and interesting.
Response: We thank Reviewer #3 for the feedback and suggestions, as they have significantly improved the quality of our manuscript. We appreciate the acknowledgment of the manuscript’s overall quality.
- I would suggest adding few lines on DNA methylation of regions other than promoters, as the effects usually differ.
Response: Thank you for the suggestion. We have included a section discussing DNA methylation in regions beyond promoters in the revised manuscript. This new version now recognizes the distinct effects of methylation in different genomic regions on transcriptional activity and gene expression. Please see page 2, lines 46-52.
- Also, both PAH and PH were used to denote pulmonary arterial hypertension, and the authors should stick to only one throughout the text.
Response: Thank you for highlighting the inconsistency in terminology. Pulmonary arterial hypertension (PAH) is a subgroup of pulmonary hypertension, and in response to the reviewer’s comments, we have opted for uniformity by consistently using “PAH” throughout the text. Except for Persistent pulmonary hypertension of newborn, lines 212-213. Additionally, we have thoroughly reviewed the entire manuscript to enhance its clarity and coherence, thereby improving the overall language quality. We believe our corrections have significantly improved the manuscript’s clarity and readability.
Round 2
Reviewer 2 Report
Comments and Suggestions for Authors
This is a review article on the possible involvement of SINA3 in PAH. However, there still exists a lot of problems in the article.
1. There are a lot of language problems in the article, so it needs in-depth review.
2. “SIN3a acts as a scaffold protein and was initially 16 described in cancer as a major regulator of epigenetics.” Should be changed to “SIN3a acts as a scaffold protein and was initially 16 described in cancer as a major epigenetic regulator”.
3. In sentence “However, activation or 20 repression of gene expression…”, “the” should be added to the place before “activation”
4. In sentence “…gene expression depends on what factors interact with SIN3a…”, “what” should be changed to “the”.
5. In sentence “This comparison offers new therapeutic possibilities”, “possibilities” should be changed to “options”.
6. For language, et al.
7. In the summary section, the overall summary is too long and not concise enough. This paper mainly discussed the relationship between SINA3 and PAH, but in the review, the author spent a lot of space describing the relationship between SINA3 and tumor. We recommend the authors to review the section of abstract.
Comments on the Quality of English Language
There are a lot of language problems in the article, so it needs in-depth review.